# A microfluidics platform for combinatorial drug screening on cancer biopsies

Federica Eduati [1,2,8], Ramesh Utharala[2], Dharanija Madhavan[2], Ulf Peter Neumann[3,4], Thomas Longerich[5,9], Thorsten Cramer [4,6], Julio Saez-Rodriguez [1,7,10] & Christoph A. Merten[2]

Screening drugs on patient biopsies from solid tumours has immense potential, but is challenging due to the small amount of available material. To address this, we present here a plug-based microfluidics platform for functional screening of drug combinations. Integrated Braille valves allow changing the plug composition on demand and enable collecting >1200 data points (56 different conditions with at least 20 replicates each) per biopsy. After deriving and validating efficient and specific drug combinations for two genetically different pancreatic cancer cell lines and xenograft mouse models, we additionally screen live cells from human solid tumours with no need for ex vivo culturing steps, and obtain highly specific sensitivity profiles. The entire workflow can be completed within 48 h at assay costs of less than US$ 150 per patient. We believe this can pave the way for rapid determination of optimal personalized cancer therapies.

[1] European Molecular Biology Laboratory, European Bioinformatics Institute (EMBL-EBI), Wellcome Trust Genome Campus, Hinxton, CB10 1SD Cambridge, United Kingdom. [2] European Molecular Biology Laboratory (EMBL), Genome Biology Unit, Meyerhofstrasse 1, 69117 Heidelberg, Germany. [3] Department of Surgery, RWTH University Hospital, 52057 Aachen, Germany. [4] ESCAM – European Surgery Center Aachen Maastricht, Aachen, Germany and Maastricht, The Netherlands. [5] Institute of Pathology, RWTH University Hospital, 52057 Aachen, Germany. [6] Molecular Tumor Biology, Department Surgery, RWTH University Hospital, 52057 Aachen, Germany. [7] Joint Research Centre for Computational Biomedicine (JRC-COMBINE), RWTH Aachen University, Faculty of Medicine, 52057 Aachen, Germany. [8] Present address: Department of Biomedical Engineering, Eindhoven University of Technology, 5600MB Eindhoven, The Netherlands. [9] Present address: Institute of Pathology, University Hospital Heidelberg, 69120 Heidelberg, Germany. [10] Present address: Institute for Computational Biomedicine, Heidelberg University, Faculty of Medicine, BIOQUANT-Center, 69120 Heidelberg, Germany. These authors contributed equally: Federica Eduati, Ramesh Utharala. Correspondence and requests for materials should be addressed to J.S-R. (email: julio.saez@bioquant.uni-heidelberg.de) or to C.A.M. (email: merten@embl.de)

Most efforts in personalized medicine have been focusing on tailoring the treatment to the specific patient based on genomic data, which are increasingly available due to advances in sequencing technologies. While there have been some impressively successful examples[1,2], cancer genomics is generally very complex and, despite the increasing knowledge on occurring mutations, there is still limited understanding on how they affect drug response[3]. Multiple efforts have been devoted to the large-scale in vitro screening of drugs across cell lines[4–6] that have proven useful to identify some genomic markers associated with drug response. However, molecular data alone has not proven sufficient to predict the efficacy[7] or toxicity[8] of a drug on an individual cell line in a reliable way. This predictability is likely to be even lower in patients, given the additional complexities when compared to cell lines.

Due to these limitations, genomics data have to be supplemented with other information in order to optimally guide the treatment for each patient, and systems for phenotypic stratification are urgently needed[3,9]. The need for new approaches is even more acute for the application of drug combinations. Combinatorial targeted therapy has been shown to be a powerful tool to overcome drug resistance mechanisms, which can be due to tumour heterogeneity, clonal selection or adaptive feedback loops[10], and seems to be a particularly promising approach for treatment of pancreatic cancer[3]. However, strategies to identify effective combinations are still in their infancy[11].

A powerful means to overcome these difficulties would be to test the drug compound directly on patient samples. Despite recent progress toward functional testing of live patient tumour cells[9,12], drug screening technologies are limited by the need of large numbers of cells[12]. Therefore, large-scale drug screening of patient tumours has been so far limited to blood tumours[13] (where a much larger amount of malignant cells is easily accessible) or requires ex vivo culturing steps[3] (e.g., patient derived cell lines, PDX models and organoids[14,15]) that require long times to grow the cells and can cause changes in the phenotype of the cells.

In contrast, microfluidic technology allows to carry out cell-based assays in tiny volumes, thus opening the way for screens on very limited material such as primary cells or patient biopsies. In line with this, microfluidics has recently been applied successfully to the testing of a few individual drugs on cancer cells[16–18]. However, these studies were based on single-aqueous phase microfluidic systems which can process only small numbers of conditions (max ~96 including replicates, typically much less). A possible solution for further scale-up is the use of droplet microfluidics[19]. In these systems aqueous droplets surrounded by an additional oil phase serve as independent reaction vessels. They have been used to screen few (2–5) conditions on cancer cells[20]; however, their scalability for larger personalized drug screens has so far been prevented by three major challenges: (i) There are hardly any approaches enabling the easy generation of chemically distinct droplets (rather than just droplets hosting different cells or different concentrations of the same drug)[19]. (ii) Synthetic small molecules can exchange between surfactant-stabilized droplets, thus prohibiting their use as independent assay vessels[21]. (iii) Strategies for fully scalable barcoding (to correlate phenotypic readout signals with chemical droplet compositions) are missing[19].

Here we present a platform that can overcome these limitations by combining two-phase microfluidics with Braille valves[22]. Our system makes use of chemically different plugs (sequential aqueous segments of nanoliter volumes spaced out by oil)[23] and enables the screening of drug combinations directly on patient biopsies. This approach requires significantly less cells compared to conventional, non-microfluidic formats and provides one to two orders of magnitude higher throughput (in terms of conditions per experiment) than existing microfluidics systems. Furthermore, we introduce a fully scalable barcoding system allowing to clearly assign all readout signals to particular plug compositions.

As a first application, we focus here on the screening of pairwise drug combinations inducing apoptosis in pancreatic cancer cells. Pancreatic cancer has a 5-year survival rate of only 6% and is genetically highly heterogeneous, hampering the use of genetic markers for stratification[24,25]. We show how our fully integrated microfluidic platform allows to perform functional screening of drug combinations from a limited amount of cells, thus enabling the prediction of optimal personalized treatments not only for pancreatic cancer cell lines, but also for primary human tumours from the clinic.

## Results

**Microfluidic platform**. Our platform (Fig. 1a) is based on a combination of plug technology[26,27] with microfluidic Braille valves[22,28] controlling individual fluid inlets of the microfluidic chip (Fig. 1a–d, Supplementary Fig. 1 and Supplementary Movie 1). To facilitate the alignment of the chip with the Braille display below, we designed additional channels that can be filled with dye (top and bottom left corner of the chip shown in Fig. 1a) and easily brought into line with particular Braille pins. All reagents (including the cell suspension, drugs and assay components) are permanently injected into the device and, depending on the valve configuration, sent either to the waste (each two inlets share one waste outlet; Fig. 1a) or to a droplet maker with a T-junction geometry. This approach allows rapid switching (~300 ms) between 16 liquid streams and, upon injection of fluorinated oil at the T junction, the generation of combinatorial plugs at high throughput. If necessary, the chemical diversity can be increased further by connecting an autosampler to one of the inlets (sequentially loading compounds from microtiter plates; Supplementary Figs 2 and 3, and Online Methods), but this procedure also requires a higher number of cells. All plug generation steps are executed in a fully automated fashion using an in-house LabVIEW program (see Online Methods).

To avoid plug breakup and/or the adhesion of plug contents to the channel walls (generally termed "wetting"—an effect that can cause significant cross contamination) we integrated an additional mineral oil inlet into our chip design, enabling the insertion of mineral oil droplets in between all (aqueous) plugs (Supplementary Movie 2). Such three-phase systems are particularly efficient in keeping plugs separated[29], even under conditions that normally cause wetting. A further crucial factor for reducing wetting was the use of special, protein-free media[27]. It is well known that protein adsorption can render Polydimethylsiloxane (PDMS) surfaces hydrophilic[30], which in turn increases the tendency of plugs wetting the channel walls. Therefore we chose FreeStyle Media (Gibco) for all experiments, as it does not contain proteins and allows the culturing of cells in absence of fetal bovine sera (FBS, which is protein-rich itself). However, while a combination of these measures significantly improved reliability, plug integrity could still not be ensured for all plugs. Hence an identification system that would still work if individual plugs break or fuse had to be implemented. Therefore, we introduced sequential barcodes to separate and identify plugs belonging to different experimental conditions, based on sequences of plugs with binary (high/low) concentrations of the blue fluorescent dye cascade blue (Fig. 1e, g and Supplementary Movie 3). This way numbers assigned to each tested condition, can be encoded (e.g., high-low = binary number "10" = decimal number "2"). Plugs were stored in PTFE tubes (Fig. 1f) and the readout was performed by flushing the plugs through the

detection module. In our case, we used an optical setup (Supplementary Fig. 4) with three different excitation lasers (375, 488 and 561 nm) and performed a multiplexed readout at three different wavelengths (450, 521 and >580 nm) to allow highly multiplexed assays in the blue, green and red spectrum,

using photomultiplier tubes (PMTs). To demonstrate the power of our barcoding approach, we converted a simplified EMBL logo (Supplementary Fig. 5) into a binary black and white image and translated all 2808 pixels into a sequence of plugs with two different fluorescence intensities. These barcodes were then

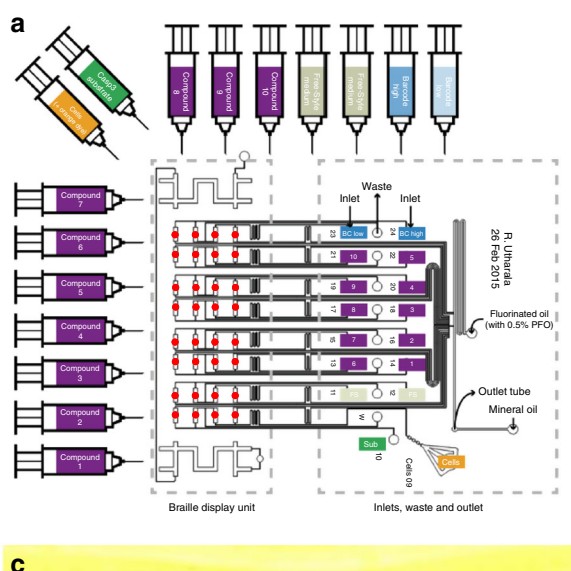

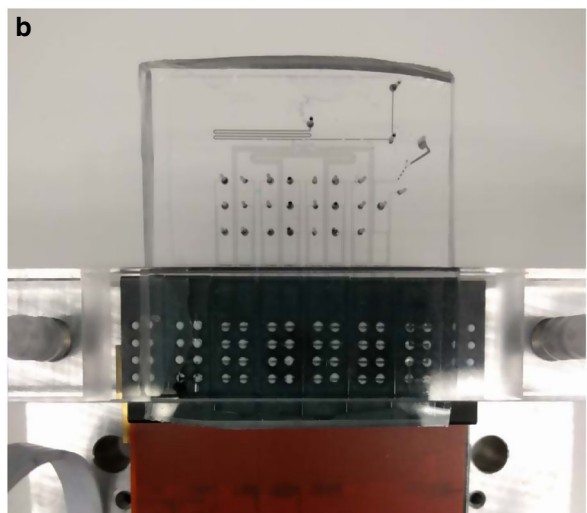

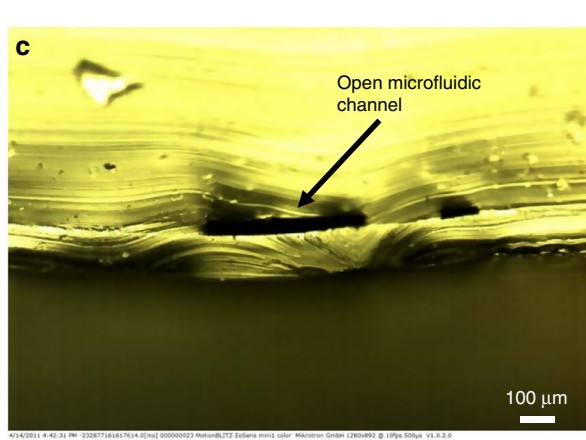

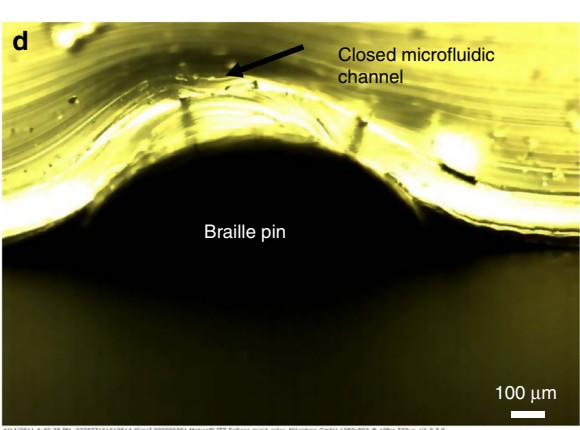

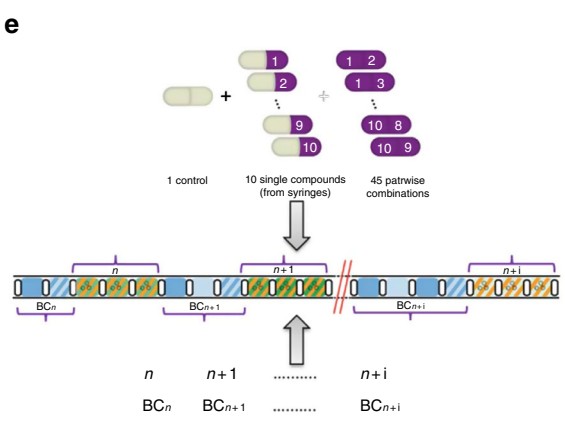

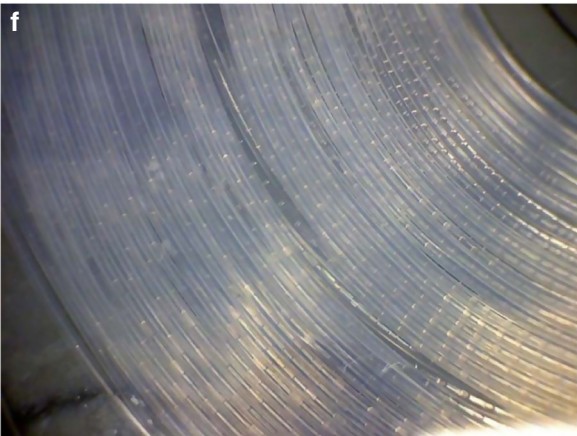

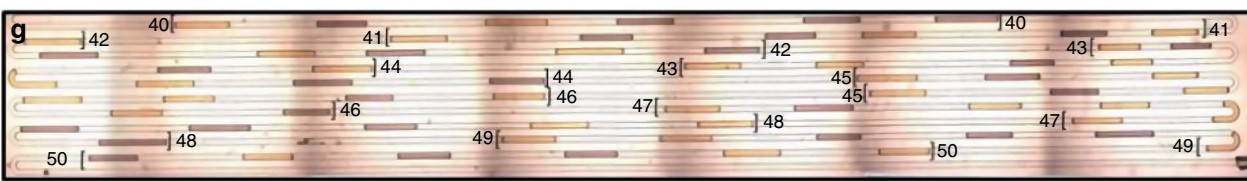

detected using our laser spectroscopy setup (Supplementary Fig. 4) and converted back into the initial image, which did not show a single mistake. This clearly demonstrates the scalability and reliability of our strategy.

**Combinatorial drug-response screening.** Motivated by the urgent need to screen therapy options directly on patient material, and by the idea that the small reaction volumes of ~0.5 μL per plug make our technology compatible with the screening of patient biopsies, we applied the above-described technology to screen cellular responses to drug combinations. Indeed, the amount of starting material (and thus viable cells) is the most limiting factor in the application of standard screening technologies directly on patient samples, and miniaturization allows screening one to two orders of magnitude more conditions using the same starting material, but simply 10x–100x smaller assay volumes. Accordingly, only ~100 cells were encapsulated in each plug together with media, one or two compounds and a rhodamine 110 (green-fluorescent dye) conjugated substrate of Caspase-3, which is an early marker of apoptosis[31]. The cell suspension contained Alexa Fluor 594 (orange-fluorescent dye) to verify its addition to all plugs. In turn, this also allowed us to monitor the correct operation of all valves since the malfunctioning of one valve would result in an unbalanced mixture of the components, thus affecting the orange signal of the corresponding plug. This is also true for the single compound conditions for which we added only media instead of the second compound.

For initial screens, we included ten well-characterized drugs and biologicals (listed in Table 1) that were tested alone and in pairwise combinations. Specifically, we included two drugs that are currently used in clinical chemotherapy as first line treatment for pancreatic cancer (Gemcitabine and Oxaliplatin), seven drugs that have specific kinase targets which play key roles in different pathways (i.e., IKK, MEK, JAK, PI3K, EGFR, AKT and PDPK1 inhibitors) and one cytokine (TNFα) that activates the extrinsic apoptosis pathway. Compounds were stored in ten syringes which were connected to the corresponding inlets in the chip (Fig. 1a).

**Table 1 List of screened compounds and their targets**

| Compound name | Compound type | Putative target (effect) |
|---|---|---|
| ACHP | Targeted | IKK (inhibition) |
| AZD6244 | Targeted | MEK (inhibition) |
| Cyt387 | Targeted | JAK (inhibition) |
| GDC0941 | Targeted | PI3K (inhibition) |
| Gefitinib | Targeted | EGFR (inhibition) |
| MK-2206 | Targeted | AKT (inhibition) |
| PHT-427 | Targeted | AKT, PDPK1 (inhibition) |
| Gemcitabine | Cytotoxic | DNA replication (inhibition) |
| Oxaliplatin | Cytotoxic | DNA replication (inhibition) |
| TNFα | Cytokine | TNFR (activation) |

Additionally we included two syringes with media to be able to generate the single-compound conditions (cells + Caspase-3 substrate + compound X + media) and the control condition (cells + Caspase-3 substrate + media + media). Plugs for different experimental conditions were generated sequentially and each condition was preceded by barcoding plugs as described in the previous paragraph. We generated 12 consecutive plugs (replicates) for each of the 55 perturbed conditions and 20 for each control condition (Fig. 2a). In order to monitor the stability of the system over time, the control condition was repeated at the beginning and at the end of the sequence, as well as after every 10 conditions. This resulted in a total of 62 sequential conditions corresponding to 800 assay plugs and 340 barcoding plugs (1140 in total). In order to rigorously assess the robustness of the results, the whole sequence of conditions (which we call run) was repeated multiple times (at least two consecutive runs for patient material and six runs, also on different days, for cell lines available in big quantities). Plugs were stored in gas permeable tubing with a total length of 5 m (having a capacity for about 3500 plugs in total) and incubated for 16 h (see details in Online Methods).

**Data extraction and quality assessment.** In the fluorescence data, each plug corresponds to a peak in one or more channels (i.e., green, orange and blue), as shown in Fig. 2b. When processing the data, we first identified the peaks in the blue channel, representing the barcodes, and we used them to separate and identify the peaks corresponding to different conditions (Online Methods). For each experimental condition we obtained multiple peaks (typically 12 replicates per run, 20 for control conditions) with signals both in the green and in the orange channel. The height of each peak is proportional to the measured fluorescence intensity: the intensity in the green channel represents the activation of Caspase-3 (thus apoptosis) while the intensity in the orange channel represents the concentration of the orange-fluorescent dye, which was added to the cell suspension. Since each plug was produced by mixing four components (cells, Caspase-3 substrate and two compounds/medium), the intensity in the orange channel allowed assessing the quality of this mixture and was used to discard conditions/peaks with extreme values (i.e., outliers, see Online Methods). The width of the peak represents the length of the plug: Thin or wide peaks were discarded as they correspond to split or fused plugs, respectively. Additionally, the first peak for each condition was discarded to avoid any possible effect of cross contamination between conditions, which could be due to the dead volume (causing a leftover of reagents from previous plug; see Supplementary Note 1) in the chip. Even if two to three peaks per condition are discarded at this step, >1200–1300 actual data points for each sample remain (even when only two runs are available). As illustrated in Fig. 2a, data were first processed separately for each run, obtaining a $z$-score and a $p$-value for each condition, and then combined to get overall scores (details in Online Methods).

**Fig. 1** Microfluidic setup. **a** Chip design. A total of 16 syringes with aqueous samples are connected to the inlets in the microfluidic chip via tubing (10 with compounds, 2 with medium to generate single drug and control conditions, 2 for barcoding, 1 for the cell suspension, 1 with Caspase-3 substrate to detect apoptosis). Other 2 inlets in the microfluidics chip are used for carrier oil (FC-40) and mineral oil. The braille display unit is used to control the valves (red coloured circles) and regulate the flow coming from the aqueous phase syringes, resulting in different combinations. Plugs are collected in a tube connected to the outlet. **b** Experimental setup. Microfluidic chip mounted onto a Braille display aligning the microfluidic channels with the braille valves. **c, d** Cross section of a valve mounted on the Braille display in the open and closed configuration. **e** Combinatorial plugs production. Single compounds and pairwise combinations are automatically generated. First barcode (BC) plugs are generated followed by the corresponding assay plugs for each condition. **f** Storage of plugs. Plugs are stored in PTFE tubing for incubation and readout purposes. **g** Array of binary barcodes in a microfluidic channel. Plugs contain two different dyes (bright colour indicating a "1", dark color indicating a "0") were used to generate binary numbers 40–50

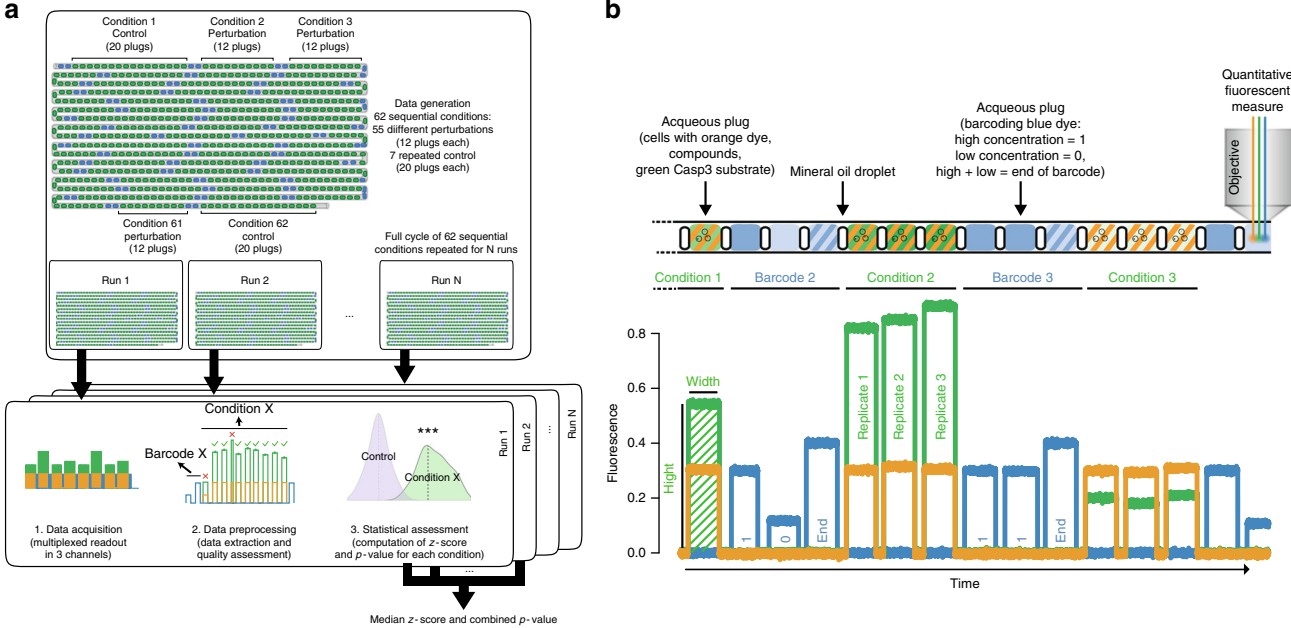

**Fig. 2** General workflow of data acquisition and processing. **a** Different conditions are generated sequentially with multiple replicates (aqueous plugs) produced for each condition and collected in a tube. The whole experiment (run) is repeated multiple times and data for each run are processed separately and then combined. **b** Example of a plug sequence and corresponding readout. Each aqueous plug is generated by mixing the following components: cells (with orange-fluorescent dye to verify the proper mixing of all components), caspase-3 substrate and one or two compounds. Plugs of each tested condition are preceded by binary barcode plugs to encode the corresponding identification number (high concentration of blue fluorescent dye = 1, low concentration = 0, followed by an end of barcode signal)

**Combinatorial screening of pancreatic cancer cell lines**. To optimize and validate our microfluidics platform, we performed combinatorial screening of drugs and biologicals on two pancreatic cancer cell lines with different genotype and phenotype[32]: AsPC1 and BxPC3. The sequence of 62 conditions was repeated six times and data from these 6 runs are shown in Fig. 3a (data refer to BxPC3 cells, but sequence of tested conditions was the same for both cell lines). These results (Fig. 3a for BxPC3 and Supplementary Fig. 6 for AsPC1) show that repeated control conditions (in purple) remained constant throughout the entire screen, confirming that cell occupancy in plugs is consistent over time, and that storing the cell suspension during plug production in the syringe on ice (protocol described in Online Methods) did not affect the viability of the cells. In order to further test cell occupancy, we produced plugs with high ($2 \times 10^5$ cells/ml, same as used in the screening experiments) and low ($0.5 \times 10^5$ cells/ml) cell densities generating a sequence of 50 plugs (replicates) for each condition in an alternating fashion (both conditions with high and low cell density were repeated 31 times for a total of 3100 plugs). By staining the cells with a fluorescent cell viability dye prior to encapsulation together with lysis buffer (to release the fluorescent dye from the cells for easy detection), the occupancy was monitored in real time for about 200 min (plug generation for screening experiments typically takes <2 h). The distribution of the median fluorescence intensity measured for the conditions with high and low cell density showed clear separation between the two conditions (t-test $p$-value $<2.2e{-}16$; Cohen's d effect size = 13.28; Supplementary Fig. 7) and no correlation with time (Pearson correlation $r = 0.09$, $p$-value = 0.61 for the cell population used in the experiments). Therefore, we could further confirm that the distribution of cells per plug is robust and stable over time throughout the duration of our experiments.

Focusing on the drug perturbation data, we found that 30 (out of the 55) conditions showed strong (z-score >0) and significant

($p$-value <0.05) effect for BxPC3 cells and 34 for AsPC1 cells (Fig. 3b, c). Among these, 19 perturbations were efficacious in both cell lines (e.g., GDC0941 and Cyt387, z-score = 0.46 for BxPC3 and 0.67 for AsPC1) while, more interestingly, some stronger effects were shown to be specific for each cell line.

**Cell line-specific drug combinations and validation**. We then focused on the cell line-specific effects, as this allows excluding drug combination that are generally toxic (which would show strong effect in both cell lines) and illustrates the concept of personalized medicine prioritizing the therapy most suited for each patient (cell line in this case). As a general observation (Fig. 3d), we noticed that PHT-427 induced apoptosis in BxPC3 cells in combination with multiple drugs (z-score range = [0.86, 2.36]; median z-score = 1.39), while it showed little or no effect on AsPC1 cells (z-score range = [−0.95,0.16]; median z-score = −0.58). For a more systematic identification of strictly cell line-specific effects, we considered the efficacious perturbations for each cell line (as described in the previous section) and assessed if the measured effect is statistically stronger than the one exerted on the other cell line (using a Wilcoxon rank-sum test). The resulting top 10 treatments which are strictly specific for each cell line (i.e., significantly stronger, with $p$-value <0.05) are shown in Fig. 3e.

The strongest effect for BxPC3 cells was measured in response to combinatorial treatment with PHT-427 (AKT and PDPK1 inhibitor acting on the PH domain) and MK-2206 (allosteric inhibitor of AKT). We compared the combinatorial effect on BxPC3 cells with the corresponding single drug treatment (Fig. 4a); either single treatment showed an effect that was not significantly higher than zero ($p$-value 0.21 and 1 for PHT-427 and MK-2206, respectively, one-tailed t-test), while the combinatorial treatment showed a strong and significant effect (effect

size = 2.24, Cohen's d; $p$-value = 0.001, one-tailed $t$-test). On the contrary, no significant effect was shown for AsPC1 cells for PHT-427, MK-2206, or their combination ($p$-value = 1, 0.68 and 0.28, respectively, one-tailed $t$-test), suggesting that the efficacy of the drug combination on BxPC3 cells is not caused by a general toxicity of the drugs when administered in combination, but rather by a cell line specific effect. Similarly, the most promising treatment specific for AsPC1 cells is the combination of Gefitinib (EGFR inhibitor) with ACHP (IKK inhibitor), which showed a strong efficacy for AsPC1 cells (effect size = 1.34, Cohen's d; $p$-value = 0.01, one-tailed $t$-test) but not for BxPC3 cells ($p$-value = 0.83, one-tailed $t$-test).

For both examples, results were validated in vitro in subsequent tissue culture experiments (Fig. 4b–d), confirming the behaviour observed in the microfluidic system. Encouraged by these results, we went on to investigate if our findings would also hold in an in vivo setting, using xenograft tumour models generated using AsPC1 and BxPC3 cell lines (see Online Methods). The BxPC3-specific combination of PHT-427 and MK-2206, as predicted from the microfluidics data, was significantly more effective than the standard of care Gemcitabine for the BxPC3 mice but not for AsPC1 mice (Fig. 5a; $p$-value 0.08, 0.03, 0.02, respectively, at day 9, 11, 14 for BxPC3; $p$-value always >0.1 for AsPC1). This specificity can be further assessed by

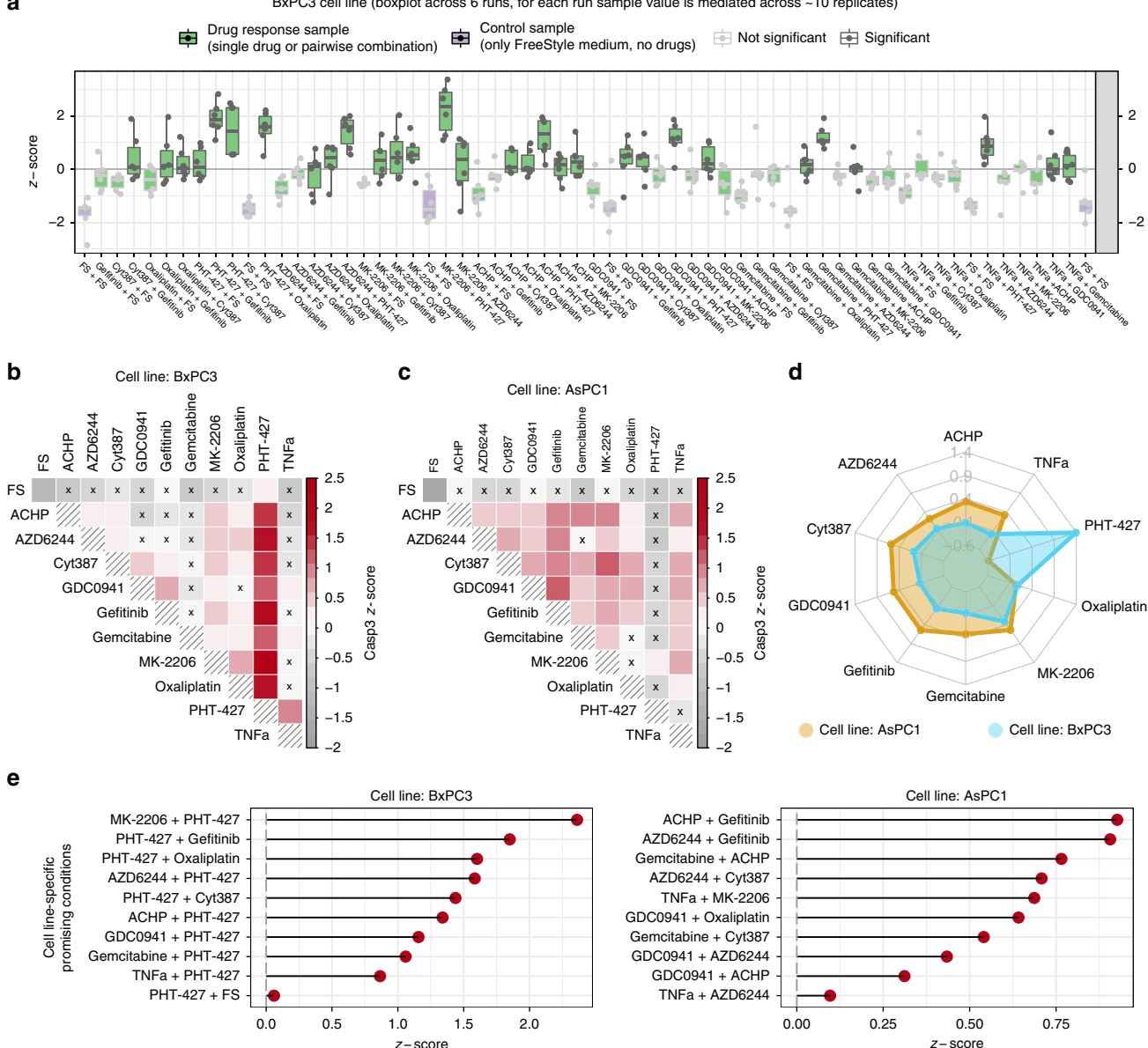

**Fig. 3** Microfluidic combinatorial drug screening in cell lines. **a** Boxplot of the sequence of conditions across multiple replicates for BxPC3 cells ($z$-scores of Caspase-3 activity) with control conditions in purple and conditions which are not significantly (Wilcoxon rank-sum test; $p$-value < 0.05) better than the control with light grey borders and dots. For each condition, replicates are shown as dots, and summary statistics are represented using a horizontal line for the median and a box for the interquartile range. The whiskers extend to the most extreme data point, which is no >1.5 times the length of the box away from the box. **b** Heatmap representation of the same conditions as in **a** (BxPC1 cell line) using median value across 6 replicates. Colored red scale starting from $z$-score equal to 0 (i.e., median activation across all conditions) while negative values are represented in grey. Conditions which are not significantly better than the control are marked with an x. **c** Heatmap representation for AsPC1 cells. **d** Overall drug efficacy for each cell line, computed as median $z$-score across all conditions (single or combinatorial perturbations) involving the drug. **e** Top 10 cell line-specific conditions for each cell line

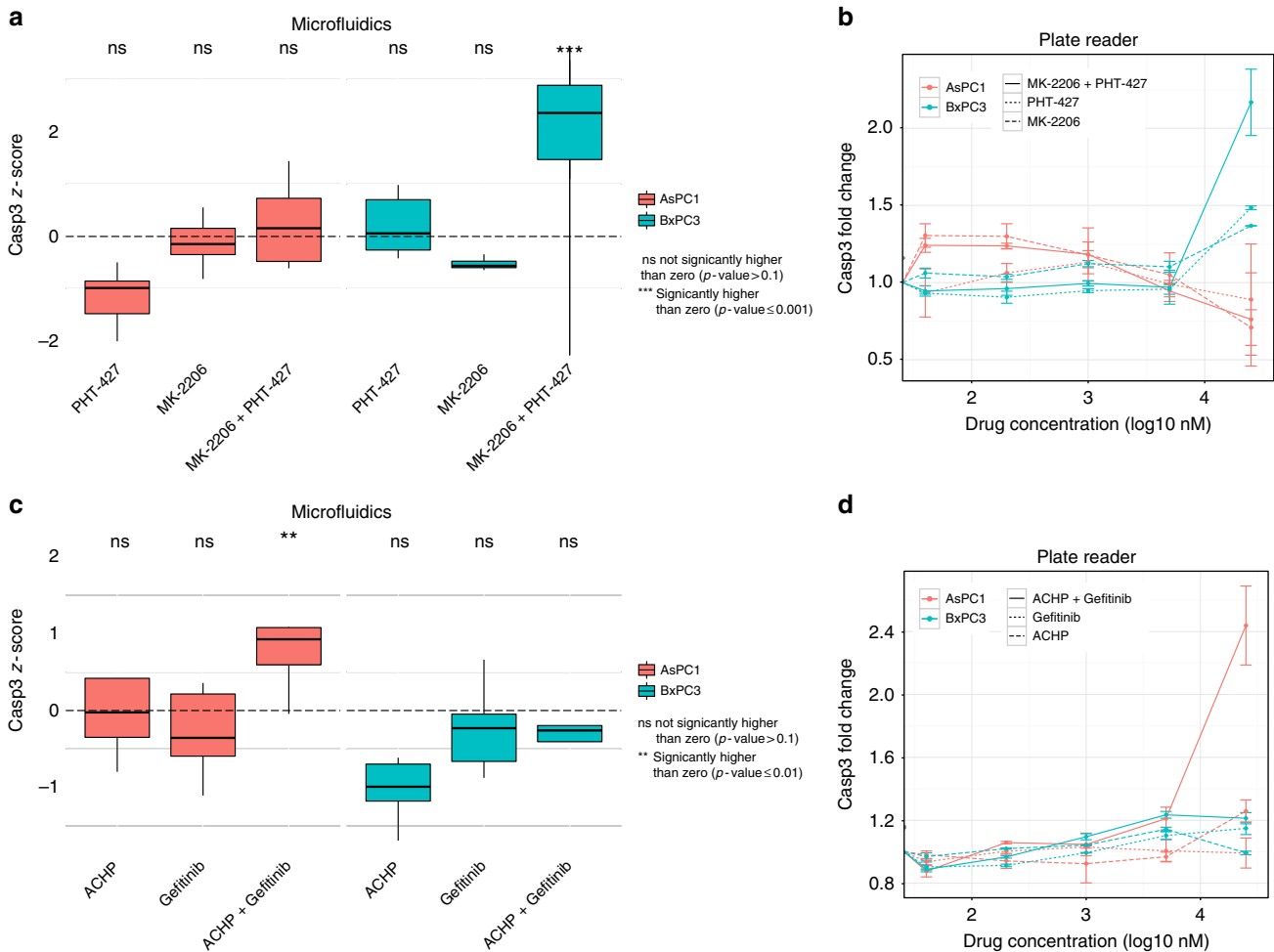

**Fig. 4** Validation of cell line-specific drug combinations in tissue culture experiments. **a** In all boxplots the horizontal line represents the median, the box the interquartile range and the whiskers extend to the most extreme data point which is no >1.5 times the length of the box away from the box. Conditions significantly higher than zero are marked (** one-tailed *t*-test; *p*-value ≤ 0.01). BxPC3 show a strong activation in the microfluidic system when treated with the combination of MK-2206 and PHT-427, which is not seen when treated with the single drugs. No strong activation is shown for AsPC1 treated with the same drugs. **b** Same behaviour is confirmed in 96-well plates where the drug combination shows a much higher signal than the two single drug samples for BxPC3 cells, while it stays at the basal level for AsPC1 cells. Error bars represent the standard error of the mean from two replicates. **c**, **d** Similarly, the combination of ACHP and Gefitinib is potent in AsPC1 cells but not in BxPC3 cells, both in microfluidic plugs (**c**) and in a 96-well plate format (**d**)

comparing the percentage of variation with respect to the untreated condition (i.e., vehicle alone; Fig. 5b), which shows that this combination is significantly more effective in the BxPC3 mice than in AsPC1 mice (*p*-value = 0.02). For the combination of Gefitinib and ACHP, predicted to be specific for AsPC1, we could only statistically compare data at day 3 due to toxicity issues of this experimental drug combination (see Online Methods). However, already at this early time point AsPC1 mice were responding better to this therapy than to Gemcitabine (*p*-value = 0.09) or to the BxPC3-specific combination of MK2206 and PHT-427 (*p*-value < 0.04; Fig. 5a top panel), while no difference was shown for BxPC3 mice. Overall, these validation experiments confirmed that our approach allows us to predict optimal (personalized) drug combinations, which indeed show a significant advantage over standard care treatments in mouse tumour models.

**Combinatorial screening of patient pancreatic tumour biopsies.** After optimizing and validating our approach in tissue culture and mouse models, we aimed for demonstrating its

applicability to clinical samples. Starting with biopsies from five patients, the resected solid tissue was dissociated to create a single cell suspension that was then used for the screening (Fig. 6a; protocol in Online Methods). Similar to the pipeline used to screen cell lines, a total of 62 conditions were screened, with 12 sequential plugs for each perturbed condition and 20 for the control conditions (each plug with about 100 live cells). The whole sequence of conditions was repeated two or three times (different runs) depending on the amount of viable cells that were obtained from each patient biopsy, with ~1 million viable cells being sufficient to perform two full runs. As in the case of the cell lines, unreliable peaks/conditions were discarded based on the intensity in the orange channel and on the length of the peaks. Only conditions passing the quality assessment in at least two runs and with consistent values across runs were considered for further analysis (boxplot with data for each run shown in Supplementary Fig. 8, more details in Online Methods). As previously described, control conditions were repeated regularly throughout the experiment and used to assess if cell occupancy and viability were robust over time. For one patient sample this

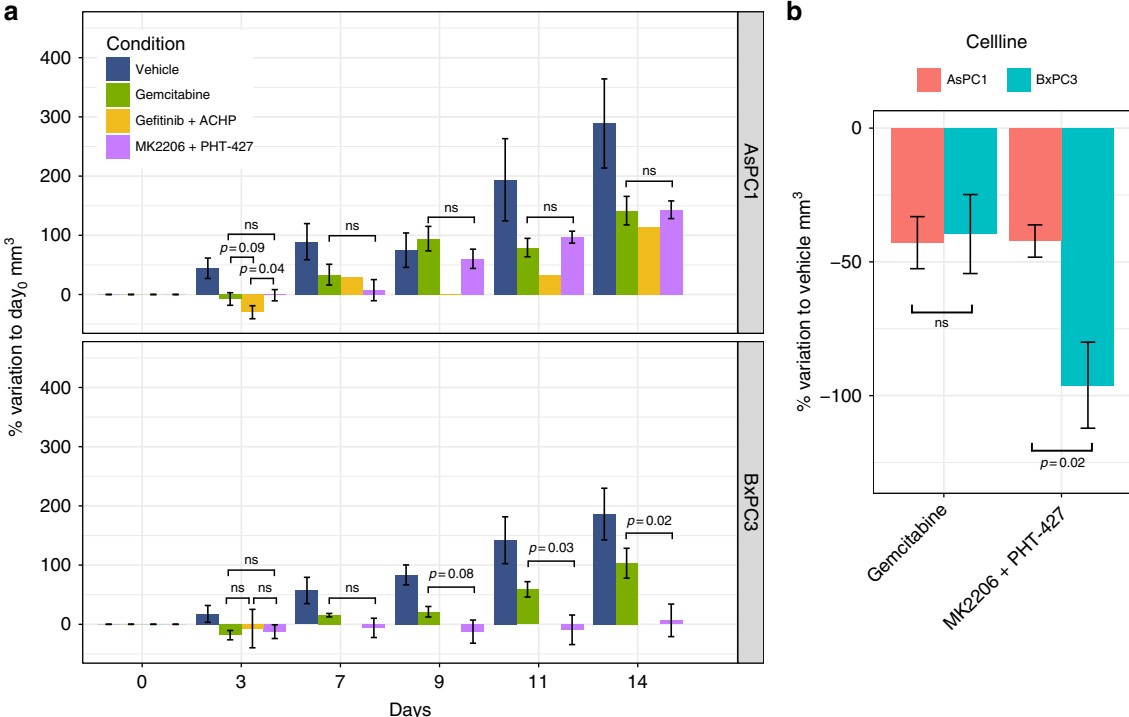

**Fig. 5** In vivo validation of drug combinations in mouse xenograft models. **a** Percentage increase of tumour size ($mm^3$) with respect to the first day of treatment (day 0) for AsPC1 and BxPC3, respectively. Statistical comparisons are reported with the corresponding *p*-value (if <0.1) or as 'ns' (not significant) when *p*-value >0.1. **b** Percentage variation with respect to the vehicle is shown to directly compare the two cell lines when treated with Gemcitabine or combination of MK2206 and PHT-427. For both panels, *p*-values are computed using one-tailed *t*-test and error bars represent the standard error of the mean. The experiment started with 5 mice per each group, error bars are not reported for the condition Gefitinib + ACHP after the third day, where only one mouse survived

condition was not fulfilled and the sample was therefore discarded from further analysis.

The four successfully screened samples (Supplementary Fig. 9) include two primary tumours from patients diagnosed with pancreatic cancer (classified as stage pT3 pN1 M0 and pT3 pNx M0, respectively) as well as a intraepithelial neoplasia (pre-tumour stage, PanIN1/2) and a liver metastasis of pancreatic cancer patient (stage pT3 pN1 M1). Heterogeneous samples were included to show that the developed technology can be used to screen different tissue types and (pre-)tumour stages.

Looking at the overall efficacy of each drug across all combinations (i.e., the median *z*-score across all conditions involving the drug; Fig. 6b), PHT-427 showed the highest efficacy for primary tumour #1 (median *z*-score = 0.75), but showed very little efficacy for primary tumour #2 (median *z*-score = 0.04), thus reproducing the strong specificity of this drug in certain pancreatic cancer samples as also observed for the cell lines. The strongest overall efficacy for primary tumour #2 was in response to Gemcitabine (median *z*-score = 0.41), which is indeed the standard of care in pancreatic cancer. Interestingly, the intraepithelial neoplasia was highly responsive to Oxaliplatin (median *z*-score = 0.72) which is also a first line treatment in pancreatic cancer. A possible explanation for this strong response is that cancer precursor lesions (PanIN) are known to be highly proliferative[33] and Oxaliplatin targets DNA replication. The strongest overall response for the liver metastasis was observed for treatment with Gefitinib (median *z*-score = 0.82) which is in line with the previously observed efficacy of EGFR inhibitors in patients with advanced pancreatic cancer[34].

Looking at the patient-specific response to combinatorial drug treatments in more detail (Fig. 6c, d) we observed that the top hit

is different for each patient (PHT-427 and Oxaliplatin for primary tumour #1, TNFa and Oxaliplatin for primary tumour #2, GDC0941 and MK-2206 for the intraepithelial neoplasia and AZD6244 and PHT-427 for liver metastasis). It is worth noticing that none of these four top combinations was effective in all samples (suggesting that they are not just generally toxic), but they were all effective in at least two samples (Fig. 6e). This suggests that they might be promising combinations for clinical applications, especially the combination of PHT-427 and Oxaliplatin, which showed strong efficacy for both primary tumour samples (*z*-score = 2.78 and 0.85, respectively). Overall, the two standard care treatments, Gemcitabine and Oxaliplatin, were recurrent across the top combinations, especially for primary tumours where Oxaliplatin is among the top hits for both patients. Another interesting observation is that both AKT inhibitors (MK-2206 and PHT-427) are effective in combination with different drugs across different patients (Fig. 6e). This supports the current interest in combinatorial treatments targeting AKT as druggable target downstream of KRAS in pancreatic cancer[35], including pre-tumour and metastasis cases[36].

## Discussion

We describe here a fully integrated microfluidic platform enabling the fast screening of many drug combinations in cell-based assays. The very small assay volumes (~100 cells per 0.5 µL plug) allowed us to perform comprehensive screens directly on patient biopsies. Importantly, these can be done without the need for any intermediate cultivation steps, which might introduce significant cellular changes and/or the selection of individual clones. Hence our approach opens the way for comprehensive screens that have

so far been restricted to blood tumours, for which patient-derived cancer cells are available in large quantities[13]. Furthermore, the low volume and the high level of automation allow such screens to be performed cost-efficiently and fast enough to adjust therapeutic treatments in a clinical setting: Within 48 h after tumour resection or biopsy, and with consumable costs of less than US$

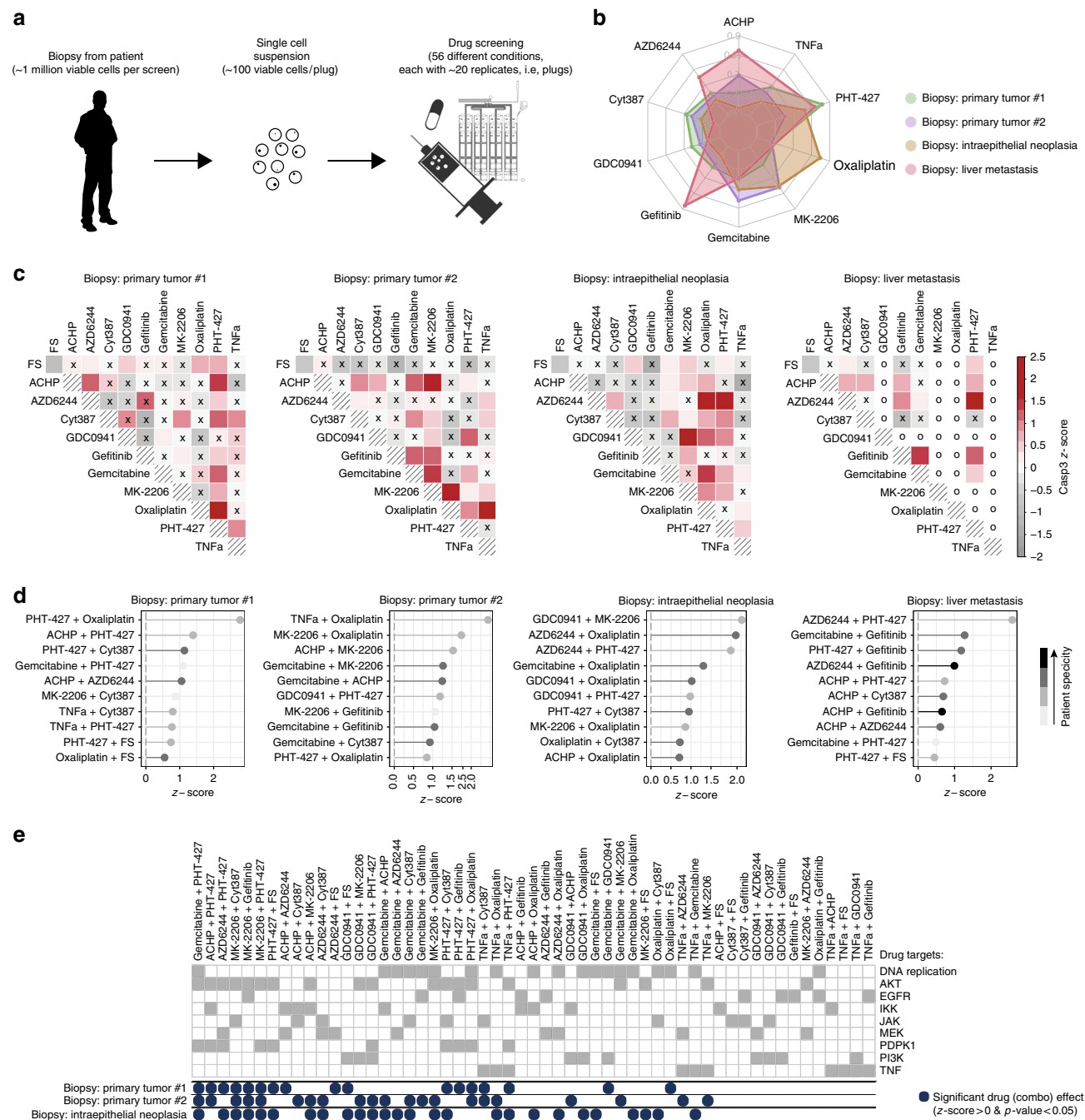

**Fig. 6** Microfluidic combinatorial drug screening of patient biopsies. **a** Workflow for patient samples. Functional drug screening on biopsies from human patients: each biopsy is dissociated to a single cell suspension, which is perturbed with different compounds using the microfluidics platform. **b** Overall drug efficacy for each cell line, computed as median z-score across all conditions (single or combinatorial perturbations) involving the drug. **c** Heatmap representation of efficacy of single and combinatorial drug perturbations for each patient using colored red scale starting from z-score equal to 0 (i.e., median activation across all conditions) while negative values are represented in grey. Conditions which are not significantly better than the control are marked with an x. The corresponding cell in the matrix is marked with 'o' in case of unmeasured conditions (two of the 10 drugs were not screened for the biopsy of liver metastasis) or technical issues (e.g., conditions with plugs showing unexpected dilutions of the orange marker dye added to the cell suspension). **d** Top 10 most effective drugs or combinations for each patient. Grey scale is used to map the specificity of each combination with light grey representing very unspecific conditions (i.e., effective in all patients where the condition was tested) and black representing highly specific conditions (effective only in one patient). **e** Comparison of efficacy of different patient samples to all perturbations with drug combinations ordered from left to right based on the number of patient samples in which they are effective. Drug targets are also shown in dark grey

150 (Supplementary Note 2), which is considerably lower than routine procedures, such as MRT scans or surgical interventions. Therefore, we envision a potential translation of this technology into clinical application, supported by our in vivo validation of best specific therapy options on mouse xenograft models.

Functional combinatorial screening has great potential in predicting personalized therapy as revealed by the data presented in this paper. Our cell line screening recapitulates the lower sensitivity of KRAS mutants (AsPC1, vs wild-type BxPC1) to PHT-427[37], (for the single-drug treatment, as well as across combinatorial treatments involving PHT-427). In addition, we could suggest and validate a novel and particularly strong and specific drug combination for BxPC3 when treated with PHT-427 and MK-2206. These two compounds are both AKT inhibitors, although they act through different sites, and PHT-427 additionally inhibits PDPK1[37]. Subsequently, similar efficacious specific combinations were suggested for each patient sample. Interestingly, no combination showed strong efficacy across all cell lines/patients suggesting that the tested treatment options are not just generally toxic but rather cell line/patient specific. This inter-patient heterogeneity in response to treatment highlights the importance of our approach to personalized medicine and is in agreement with previous findings[3], where sensitivity profiles were shown to be particularly patient specific for pancreatic cancer.

A network-based perspective can be informative when investigating the best combinatorial therapies[38]. For example, mapping the effect of the kinase specific inhibitors on the signalling pathways can help to prevent the mechanisms of drug resistance by suggesting combinatorial therapies that act on parallel pathways. An interesting example is the combination of PHT-427 (AKT inhibitor) and AZD6244 (MEK inhibitor), which was highly effective ($z$-score >0.35) in three out of the four tested patient samples. These drugs in fact block two important parallel pathways which are the MEK-ERK and the PI3K-AKT pathways. Although, as far as we know, synergistic combinations targeting these pathways have been previously studied for pancreatic cancer, especially in the context of KRAS mutants (>90% of pancreatic cancer patients are KRAS mutants)[35], this is the first evidence of potential synergistic effect between PHT-427 and AZD6244. The efficacy of this treatment also in BxPC3 cells (KRAS wild type) might suggest a more general applicability.

In this study, we prioritized the number of tested drug combination over the possibility of testing multiple time points or multiple drug concentrations; although, we are aware that they are both important aspects that can affect cellular drug response. The choice of maximizing the number of tested combinations, rather than testing multiple time points, is in line with other recent studies focusing on large-scale screening[4–6], despite the risk of missing drug combinations showing only slow effects (false negatives). Similarly screening a single, relatively low concentration only implies the risk of false negatives, but should never result in false positive hits, especially when focusing on the effects that are patient/cell line-specific rather than generally toxic. Therefore, we believe that our setup is already informative for the clinical community as it allows patient-specific prioritization of therapeutic strategies among a set of approved (or under trial) alternative. Nonetheless, the extension to time- and dose-dependent results is an important focus for future work.

Compared to other personalized approaches our procedure has specific advantages: While organoids and xenografts are particularly well-suited for mimicking three-dimensional tumour architecture and the in vivo microenvironment, respectively, the use of individualized tumour cells as shown here facilitates rapid, massively parallelized assays at low cost. In order to ensure a very physiological sample composition, we deliberately decided to avoid cell sorting based on specific markers (e.g. by FACS) and to avoid any cultivation step ahead of the screen. A similar approach has also been used by Montero and colleagues[12] and is supported by the fact that there are many publications showing that tumour heterogeneity plays an important role and, in particular, stroma cells have a direct impact on the drug response[39,40]. The presented technology could be exploited further by implementing single-cell droplet assays[19,26,41], which require fewer cells and also allow to investigate intratumoral heterogeneity. In addition to deriving an averaged readout over tens to hundreds of single-cell replicates per treatment option, one could quantitatively determine the number of non-responding cells for each drug cocktail. Such data would probably be very valuable to overcome therapy resistances.

Apart from using it for personalized cancer therapies, our microfluidic platform should also be of interest for further applications, such as the screening of cocktails for targeted stem cell differentiation[42,43] or combinatorial chemistry[44]. In addition, the possibility of controlling the flow in individual channels of a network should facilitate the construction of cheap and versatile multi-way cell- and droplet-sorters. This should pave the way for further significant developments on the technology side.

## Methods

**Microfluidic setup.** Microfluidic chips were fabricated using standard soft-lithography methods. In brief, molds were fabricated on 4-inch silicon wafers (Siltronix) using AZ-40XT positive photoresist (Microchemicals) according to the manufacturer's instructions. Patterning was achieved by projecting 25400 dpi photomasks (Selba) onto the photoresist using light with a wavelength of 375 nm (Suess MicroTec MJB3 Mask Aligner). All channels had a height of ~50 µm and widths ranging from 50 to 400 µm (400 µm for all valve sections). PDMS chips were manufactured using elastomer and curing agent in a ratio of 1:10 (Sylgard 186 elastomer kit, Dow Corning Inc) cured overnight at 65 °C. To allow for valve actuation (compression of the valve sections by the Braille pins) the PDMS chips were not bonded to glass, but rather to a thin elastic PDMS membrane. This membrane was prepared using elastomer and curing agent in a 1:20 ratio, poured on a transparency sheet and spin coated at 700 rpm (Laurell WS 650) before overnight curing. Bonding was performed in a Diener Femto Plasma Oven. Connections to inlets and outlets were punched using 0.75 mm Harris Unicore Biopsy punches. Before screening, chips were flushed with Aquapel (PPG industries) from the outlet up to the T junction to render the channel surface hydrophobic.

For screening, the Braille valve chip (Fig. 1a) was mounted onto an SC-9 Braille display (KGS Corporation, Japan) using an in-house holder as shown in Fig. 1b. This holder includes a Plexiglas bar with built-in screws to push the PDMS chip onto the Braille pins. The design of the chip ensures that all fluid connections (inlets and outlets) are outside the area covered by the Plexiglas bar. Movement of the Braille pins was controlled by an in-house LabVIEW program (available for free academic download at www.merten.embl.de/downloads.html), enabling to actuate all individual pins according to a pre-defined sequence (corresponding to systematic drug combinations and barcodes). Barcodes were generated using binary concentrations of Cascade Blue (16 µM and 48 µM for low and high, respectively), as detailed in Supplementary Note 3.

For all experiments the aqueous fluids were injected at a flow rate of 500 µl/h using Harvard Apparatus Syringe pumps. Based on the continuous injection mode the ab/adsorption of small molecules in PDMS is negligible as further detailed in Supplementary Note 4. AFC-40 containing 0.5% perfluoro-octonol (PFO, ABCR) and mineral oil were injected at rates of 200 µl/h and 175 µl/h. Resulting aqueous plugs were 1.5–2 mm in length and 424–560 nL in volume.

The sequential generation of the 1140 plugs required ~1 h (3 s per plug). Plugs were incubated for 16 h at 37 °C in a 5% $CO_2$ atmosphere. The incubation time was chosen based on observations from kinetic experiments (in plate reader format), showing that robust results with good signal to noise ratios can be obtained for an overnight workflow.

Subsequent to incubation, the fluorescence of all plugs was determined inside PTFE tubing (Adtech) with an inner diameter of 600 µm using the optical setup shown in Supplementary Fig. 4. The flow rate used for the sequential measurement of the plugs was adjusted to match the time required for plug generation. In this way we could guarantee the same incubation time for each plug. The resulting data were analyzed using custom R-scripts (see paragraph "data processing" for details).

**Setup of the fluidic system and choice of additives and oils.** We used fluorinated oil without stabilizing surfactant as a carrier phase (only 0.5% of the anti-wetting agent PFO was added), which turned out to have two major advantages: upon reaching the outlet, small droplets generated at the T-junction fused and formed larger plugs that completely filled the collection tubing, thus allowing to

incubate all conditions in a sequential order. Furthermore the lack of surfactant prevented the formation of micelles, which can cause the exchange of reagents between droplets[21] (see Supplementary Note 5 for details). To increase stability of the arrays, plugs were furthermore spaced out using mineral oil (Sigma).

**Integration of an autosampler**. To allow for upscaling of the screens, we integrated an autosampler into our microfluidic platform[27]. One of the inlets of the microfluidics chip was connected to a Dionex 3000SL Autosampler, aspirating samples from 96-well plates and injecting them sequentially into a target tubing connected to an external Harvard Apparatus Syringe pump. While the resulting throughput for loading compounds from different wells is rather low (~90 s per reagent), each of them can be mixed with all of the drugs injected directly into the Braille valve chip. Therefore, the maximal throughput in terms of pairwise combinations is much higher (e.g., 9 s per combination when mixing with 10 further reagents directly connected to the Braille display chip). However, one effect had to be taken into account: the concentration of compounds coming from the autosampler varied due to dispersion of the samples in the miscible carrier phase (the buffer used in the robotic system)[45]. To overcome this effect, we implemented a feedback loop between the autosampler and the Braille valves: The relay signal of the autosampler was used to send the beginning and end of each sample from the microtiter plates to the waste, using the two Braille valves controlling the relevant inlet on the microfluidic chip. This process allowed to overcome sample dispersion and is illustrated in Supplementary Fig. 2. This way only the centre part of each sample segment, showing constant concentration, was mixed with the drugs injected directly into the Braille display chip. To verify this procedure we mixed fluorophores stored in a 96-well plate with fluorophores injected directly into the autosampler and measured the resulting fluorescence signal of the droplets (Supplementary Fig. 3). All combinations showed the expected signal intensities, confirming the feasibility of the approach if a sufficient amount of cells is available for screening.

**Single cell suspension from pancreatic primary tumours**. Primary pancreatic tumours were obtained from routine resections from patients who signed an informed consent approved by the Research Ethics Committee of the Medical Faculty of the RWTH Aachen University (EK 206/09). The project was also approved by the EMBL Bioethics Internal Advisory Committee. Informed consent was obtained from the patients for research use of the samples. A viable single cell suspension was prepared from the fresh tumours and directly used in an experiment within the next few hours. Tumours were first mechanically dissociated (~2–3 mm pieces) and digested for 1.5 h at 37 °C in 5 ml of prewarmed digestion media consisting of 1 mg/ml collagenase (Sigma) solution in DMEM/F12 (Gibco, Life Technologies). Solution was pipetted every 30 min to facilitate dissociation, diluted in 25 ml of buffer (PSB) to stop the reaction, and centrifuged at 250 g for 5 min. Supernatant was then removed and 2 ml of 0.05% trypsin-EDTA (Gibco, Life Technologies) were added and the solution was incubated for 5 more minutes at 37 °C. Subsequently 15 ml of DMEM supplemented with 10% fetal bovine serum were added and the solution was centrifuged again for 5 min at 250 g. After removing the supernatant, the pellet was resuspended in FreeStyle medium (ThermoFisher).

**Preparation of cells for microfluidic experiments**. AsPC1 and BxPC3 cell lines were obtained from ATCC for this study and were tested negative for mycoplasma contamination using the MycoAlert[TM] PLUS Mycoplasma Detection Kit from Lonza. Both cell lines were cultured using the recommended media RPMI-1640 (L-glutamine, 25 mM HEPES) with 10% FBS, 1% Sodium Pyruvate, 10% P/S and 4500 mg/l glucose. After preparing single cell suspensions from tumours (as described above) or from cell cultures (by trypsinizing, harvesting and washing the cells in PBS) they were prepared for the microfluidic experiment in the same way: Cells suspended in FreeStyle medium were supplemented with 1 mg/ml Xanthan Gum (Sigma), for density matching, and 2 µl/ml of 10% Pluronic (Sigma), to reduce cell attachment and formation of clumps. Cells were filtered using a 40 µm cell strainer and diluted to a final concentration of $8 \times 10^5$ viable cells/ml (counted with trypan blue exclusion method using a BioRad cell counter). Quantity of 15 µg/ml of the orange-fluorescent dye Alexa Fluor 594 (ThermoFisher, #A33082) were added to the cell suspension in order to verify the proper mixture of the components in the plug. Cell suspension was then pipetted in a 5 ml syringe with the tubing directly connected to the syringe (no needle to avoid clogging) using PDMS and UV glue, and with a magnetic stir bar inside the syringe. For the duration of the experiment cells were maintained at low temperature using ice and constantly stirred.

**Preparation of drugs and Caspase-3 substrate**. Cyt387 (#S2219), PHT-427 (#S1556), MK-2206 (#S1078), GDC0941 (#S1065), Gefitinib (#S1025), Oxaliplatin (#S1224), AZD6244 (#S1008) and Gemcitabine (#S1149) were purchased from Selleckchem. ACHP (#4547) was purchased from Tocris. All compounds were diluted in DMSO to a 20 mM stock solution. When preparing the syringes for the microfluidic experiment, compounds were further diluted in FreeStyle medium to a final concentration of 5 µM in the plugs. Tumour Necrosis Factor-α (TNF) (#PHC3015) was purchased from Life Technologies and diluted according to the manufacturer's instruction to a 10 µg/ml stock solution. It was further diluted in FreeStyle medium for microfluidic experiments to a final concentration of 5 ng/ml.

For validation experiments in 96 well plates, compounds were instead prepared in 5 consecutive fivefold dilutions (25, 5, 1, 0.2, 0.004 µM).

The caspase-3 substrate (Z-DEVD)2-R110 was purchased from Biomol (#ABD-13430). Volume of 3 ml of substrate working solution were prepared by adding: 2400 µl 5X Reaction buffer (20 mL of 50 mM PIPES, pH 7.4, 10 mM EDTA, 0.5% CHAPS), 60 µl DTT (1 M), 540 µl dH2O and 44 µl Z-DEVD–R110 substrate (5 mM). Caspase-3 activity is a marker of apoptosis and is a good early marker of cell viability (Supplementary Fig. 10).

**Data processing**. Data were acquired using an in-house LabVIEW program (available for free academic download at www.merten.embl.de/downloads.html) allowing fluorescence detection in three separate channels (i.e. blue = fluorescence barcodes, green = Caspase-3 activity and orange marker dye to monitor mixing of reagents) as exemplified in Fig. 2b. Peaks in each of the three channels were identified by defining an empirical threshold both on the intensity and the duration of the measured signal in order to distinguish real peaks from background noise. Peaks in the blue channel (corresponding to the barcode plugs) were detected and used to separate the different conditions consisting of sequences of peaks (replicates) in the green channel. Conditions with multiple peaks having either very low or very high orange signals (e.g., due to occasional cell clogging) were manually discarded. Additionally, we also discarded peaks showing very high and very low width (based on empirical thresholds) in order to remove peaks corresponding to fused or split plugs, respectively. After these steps, we considered the distribution of the intensity of the orange peaks across all samples and discarded the extreme values (i.e. the outliers). Where Q1 is the 25th percentile, Q3 is the 75th percentile and IQR is the interquantile range (Q3–Q1), outliers were defined as values lower than Q1 $-1.5 \times$ IQR, or higher than Q3 $+1.5 \times$ IQR. These strict rules were applied to guarantee higher quality of the data used for the analysis described in the paper. For each run we computed median value for each condition across replicates. In order to compare different runs, we then computed the z-score for each run (i.e., standardization by subtracting the mean and dividing by the standard deviation) and then computed the median across runs. We also computed the significance of the efficacy of drug treatments: p-values were computed using Wilcoxon rank-sum test to verify whether the Caspase-3 response to the tested drug combinations is significantly higher than the one measured in the control condition (no perturbation). P-values were computed separately for each run, FDR-corrected for multiple hypotheses testing and combined across different runs using Fisher's method. Significance and z-score for all data shown in the manuscript is illustrated in Supplementary Fig. 11.

**Xenograft mouse models**. Xenograft mouse experiments were performed by an external company (EPO Berlin). All animal experiments were carried out in accordance to the German Animal Welfare Act as well as the UKCCCR (United Kingdom Coordinating Committee on Cancer Research). The respective pancreas carcinoma cell suspensions of the human AsPC1 or BxPC3 cells were injected subcutaneously (s.c.) into the left flank of anaesthetized female Rj:NMRI-Foxn1 nu/nu mice from Janvier. The age of the animals was 6–8 weeks. Tumours were allowed to establish a palpable size (about 0.1 cm$^2$), before the treatment was started. A total of 40 mice were used, grouping the animals in four groups for each cell line (five animals per group). Mice in the four groups were treated respectively with: 1. Gemcitabine (100 mg/kg i.p. once a week in PBS); 2. combination of Gefitinib (75 mg/kg p.o. sequence days 0, 2, 4, 6, 8, 10, 12 in 0.5% Polysorbate) and ACHP (8 mg/kg i.v. q4d—day 0, 4, 8—in 0.9% NaCl); 3. combination of MK2206 (120 mg/kg p.o. sequence days 0, 2, 4, 7, 9, 11 in 30% Captisol) and PHT-427 (200 mg/kg p.o. twice a day in 40–50 mg/ml Sesamoil); 4. vehicle p.o. alone. During the study tumour volumes were measured in two dimensions with a caliper. Tumour volumes (TV) were calculated by the formula: TV = (width$^2$ × length) × 0.5. During the study mice were maintained under sterile and controlled conditions (22 °C, 50% relative humidity, 12 h light–dark cycle, autoclaved food and bedding, acidified drinking water) and monitored for body weight and health condition. Treatment with MK2206 + PHT-427 was interrupted for both AsPC1 and BxPC (at day 5 and 8, respectively) due to massive body weight loss for AsPC1 mice, and restarted at day 11. Treatment with Gefitinib + ACHP was stopped at day 4 for all mice due to toxicity (only one AsPC1 mouse survived until the end of the experiment). Percentage variation was computed with respect to day 0 (i.e., first day of treatment) for Fig. 5a and with respect to the vehicle for Fig. 5b Student's t-test (one-sided) was used for statistical comparison of the different groups.

**Code availability**. The code used to analyse the data are available as an R package in the GitHub page of the Saez Lab (https://github.com/saezlab/BraDiPluS). Microfluidic control software and chip designs can be downloaded from the EMBL server (www.merten.embl.de/downloads.html).

**Data availability**. The microfluidics-based drug screening data referenced during the study are available in Zenodo https://doi.org/10.5281/zenodo.1248886[46]. The authors declare that all other data supporting the findings of this study are available within the article and its Supplementary Information Files and from the corresponding author upon reasonable request.

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

## Acknowledgements

We gratefully acknowledge all patients who kindly gave their consent for research use of the biopsies. We thank Ilka Sauer, Anne Esser and Ellen Krott for highly appreciated technical assistance, EMBL electronic workshop for building Braille controllers and implementing a LabVIEW command list, EMBL mechanical workshop for building customized parts for the Braille display, Felix Krüger and John P. Overington for help selecting kinase specific inhibitors, Dominic Eicher and Nirupama Ramanathan for constructive discussion and input on the optimization of the microfluidics platform for drug screening on cells, Jan Korbel and Denes Turei for critical reading of the manuscript. F.E. thanks the European Molecular Biology Laboratory Interdisciplinary Post-Docs (EMBL EIPOD) and Marie Curie Actions (COFUND) for funding.

## Author contributions

F.E. designed and performed experiments, performed the computational analysis under the supervision of J.S.-R. and shaped the overall study design. R.U. designed and performed experiments and developed the microfluidic platform under the supervision of C. A.M. D.M. performed experiments. U.P.N. supervised the surgical specimen collection and provided clinical patient data. T.C. provided clinical samples and contributed to the design of the study. T.L. provided patho-histological patient data. J.S.-R. and C.A.M. conceived the project, designed experiments and supervised the study. F.E., J.S.R. and C. A.M. wrote the manuscript. F.E., R.U., D.M., T.C., J.S.-R., C.A.M. interpreted the results and contributed to manuscript development. All authors approved the final manuscript.
