## [Peer Review File · Nature Communications]

REVIEWERS' COMMENTS:

Reviewer #1 (Remarks to the Author):

The authors have appropriately addressed my comments, as well as other reviewers' comments. The change in tone and emphasis along with additional new data have strengthened the manuscript, which is will be of broad interest.

Reviewer #2 (Remarks to the Author):

I checked the manuscript and the response. I believe this work is almost ready to be published in Nat Comm.

Two major suggestions/comments:

1. The concept of 'wetting issue' in page 5 paragraph 3 is not common for non-expertise, please further explain it in the text.
2. The impact concerning cell purity of cancer samples in page 11 paragraph 2, which is also raised by REVIEWER 1, is not explained in the manuscript, please address this issue.

Reviewer #3 (Remarks to the Author):

I appreciate the revisions made by the authors. While limited, I also appreciate the extent of work involved in even the 2 way comparison of in vivo models they performed. I did not mean to suggest that they do new human clinical trials in my original comments. That is a great goal, but one that I would expect to be well beyond the scope of this paper.

Reviewer #4 (Remarks to the Author):

The authors revised the manuscript very thoroughly and responded to the comments in great detail. I appreciate very much their efforts on providing more details on the method and the additional data obtained with mouse xenograft models. It is now a very comprehensive study with excellent results, still on an academic level, but appropriate for publication as it is.

There are no further comments from my side.

RESPONSE TO REVIEWERS' COMMENTS:

Reviewer #1 (Remarks to the Author):

The authors have appropriately addressed my comments, as well as other reviewers' comments. The change in tone and emphasis along with additional new data have strengthened the manuscript, which is will be of broad interest.

We very much appreciate this positive conclusion.

Reviewer #2 (Remarks to the Author):

I checked the manuscript and the response. I believe this work is almost ready to be published in Nat Comm. Two major suggestions/comments:

1. The concept of 'wetting issue' in page 5 paragraph 3 is not common for non-expertise, please further explain it in the text.

In the final version of the manuscript we have added a few more explanations about wetting for readers from outside the field of microfluidics:

“To avoid plug breakup and/or the adhesion of plug contents to the channel walls (generally termed “wetting” – an effect that can cause significant cross contamination) we integrated an additional mineral oil inlet into our chip design, enabling the insertion of mineral oil droplets in between all (aqueous) plugs (Supplementary Movie 2).”

2. The impact concerning cell purity of cancer samples in page 11 paragraph 2, which is also raised by REVIEWER 1, is not explained in the manuscript, please address this issue.

We believe this point had already been fully clarified during the last revision round (reply to Reviewer 1 comment 1), during which we added further explanations on the cell purity in the discussion part. Please find below the previous reply:

Processed samples were carefully selected by the pathologist from the resected tissue to guarantee that only the specific tissue of interest was screened. However, we agree with the reviewer that this is an important point and we now include the following paragraph in the Discussion:

“In order to ensure a very physiological sample composition, we deliberately decided to avoid cell sorting based on specific markers (e.g. by FACS) and to avoid any cultivation step ahead of the screen. A similar approach has also been used by Montero and colleagues¹² and is supported by the fact that there are many publications showing that tumor heterogeneity plays an important role and, in particular, stroma cells have a direct impact on the drug response^{39,40}.”

To address the reviewer concern on the variability of the composition of cells among plugs, we would like to point out that having a large number of replicates per condition (at least 20) with a low number of measured cells per replicate (about 100) we have information not only on the average response of the cells in bulk, but also on the biological variability of the response. When defining promising drug combinations we consider only those which are consistently and robustly

effective across replicates. To corroborate the robustness of the results shown in the paper we computed also the significance of the efficacy of drug treatments. This is now applied in the data analysis pipeline (results shown in new version of Fig. 3, for cell lines, and Fig. 6, for patient samples) and described in the Online Methods:

“For each run we computed median value for each condition across replicates. In order to compare different runs, we then computed the z-score for each run (i.e. standardization by subtracting the mean and dividing by the standard deviation) and then computed the median across runs. We also computed the significance of the efficacy of drug treatments: p-values were computed using Wilcoxon rank sum test to verify if the Caspase-3 response to the tested drug combinations is significantly higher than the one measured in the control condition (no perturbation). P-values were computed separately for each run, FDR-corrected for multiple hypotheses testing and combined across different runs using Fisher’s method. Significance and z-score for all data shown in the manuscript is illustrated in new Supplementary Fig. 11.”

In this way we can be sure to prioritize patient-specific drug combinations which are statistically robust, thus effective across replicates regardless of the potentially unequal distribution of cell types (present due to tumor heterogeneity) in the plugs. New Supplementary Fig. 11 also highlights that most conditions with z-score>0 (threshold used in the previous version of the paper to assess drug efficacy) also have a FDR-corrected p-value < 0.1.

Reviewer #3 (Remarks to the Author):

I appreciate the revisions made by the authors. While limited, I also appreciate the extent of work involved in even the 2 way comparison of in vivo models they performed. I did not mean to suggest that they do new human clinical trials in my original comments. That is a great goal, but one that I would expect to be well beyond the scope of this paper.

We appreciate this positive feedback and feel encouraged to work on further translational efforts exceeding the scope of this paper.

Reviewer #4 (Remarks to the Author):

The authors revised the manuscript very thoroughly and responded to the comments in great detail. I appreciate very much their efforts on providing more details on the method and the additional data obtained with mouse xenograft models. It is now a very comprehensive study with excellent results, still on an academic level, but appropriate for publication as it is.

There are no further comments from my side.

We are very grateful for these highly positive comments.